# School Teachers’ Self-Reported Fear and Risk Perception during the COVID-19 Pandemic—A Nationwide Survey in Germany

**DOI:** 10.3390/ijerph18179218

**Published:** 2021-09-01

**Authors:** Stefanie Weinert, Anja Thronicke, Maximilian Hinse, Friedemann Schad, Harald Matthes

**Affiliations:** 1Institute of Social Medicine, Epidemiology and Health Economics, Charité-Universitätsmedizin Berlin, Corporate Member of Freie Universität Berlin, Humboldt-Universität zu Berlin, Berlin, Germany; Anja.thronicke@havelhoehe.de (A.T.); Maximilian.hinse@charite.de (M.H.); harald.matthes@charite.de (H.M.); 2Research Institute Havelhöhe (FIH), Department Network Oncology, Kladower Damm 221, Berlin, Germany; Friedemann.schad@havelhoehe.de; 3Interdisciplinary Oncology and Palliative Care, Hospital Gemeinschaftskrankenhaus Havelhöhe, Berlin, Germany; 4Medical Clinic for Gastroenterology, Infectiology and Rheumatology CBF, Charité-Universitätsmedizin Berlin, Corporate Member of Freie Universität Berlin, Humboldt-Universität zu Berlin, Berlin, Germany

**Keywords:** COVID-19, vaccination intention, fear of infection, teacher, schools, survey

## Abstract

With the coronavirus disease 2019 (COVID-19) cases peaking and health systems reaching their limits in winter 2020/21, schools remained closed in many countries. To better understand teachers’ risk perception, we conducted a survey in Germany. Participants were recruited through various associations and invited to take part in a cross-sectional COVID-19-specific online survey. Descriptive statistical analysis was performed. Factors associated with teachers’ fears of contracting the severe acute respiratory syndrome coronavirus type 2 (SARS-CoV-2) were evaluated with an adjusted multivariable regression analysis. The median age of the 6753 participating teachers was 43 years, and 77% were female. Most teachers worked in high schools (29%) and elementary schools (26%). The majority of participants (73%) feared contracting SARS-CoV-2 at school, while 77% intended to receive their COVID-19 vaccination. Ninety-eight percent considered students to pose the greatest risk. Female and younger teachers were significantly more anxious to get infected and teachers who opposed the re-opening of schools had significantly higher odds of being more anxious (*p* < 0.001). To the authors’ knowledge, this is the first study to describe teachers’ risk perception of COVID-19 and their attitudes towards vaccinations in a nationwide survey. The anxiety correlates with the COVID-19 protection measures demanded and appears to be a driving factor rather than rational logic.

## 1. Introduction

In the pandemic era of the coronavirus disease 2019 (COVID-19), the scientific interdisciplinary discourse on various coping strategies has largely been conducted by policymakers and their advisors. This has resulted in many opinions and views that often differ from scientific facts. In fact, so far, more than 3 million people in Germany have tested positively for SARS-CoV-2, which causes COVID-19. Nearly 80,000 deaths are linked to COVID-19 (as of 14 April 2021, [1]).

Prior to the launch of the vaccination campaign in November 2020, 37% of Germany’s population intended to get vaccinated [2]. With the events of the second and third waves and with increasingly stricter COVID-19 measures in place, the proportions of the population with vaccination intentions rose to 59% in February and 75% in May 2021. However, these are overall numbers that hardly mirror the vaccination intentions of different occupational groups that may have an above-average risk of SARS-CoV-2 infections. Intensive care healthcare workers’ vaccination intentions had risen from 65% to more than 75% between December 2020 and February 2021 [3]. A survey conducted in Saxony, Germany at the same time showed that a staggering 90% of teachers from public schools and around 80% from private schools intended to receive COVID-19 vaccinations [4]. The reasons for this and whether this applies nationwide are unknown. Healthcare workers are clearly at a particularly high risk of contracting SARS-CoV-2 through physical contact with infected people. Teachers are also at high risk. However, as classroom teaching has been largely reduced and COVID-19 precautions have been implemented in schools, most teachers are protected to a certain extent. The actual risk of teachers contracting SARS-CoV-2 despite COVID-19 precautions remains controversial.

Children and adolescents likely spread the infection at similar rates [5]. While at the beginning of the pandemic, studies observed no evidence of secondary transmission of COVID-19 from children attending school [6], it is now clear that infections have been imported into schools from the community, where it can spread [7]. However, further transmission has been rare when rigorous measures, such as wearing face masks and frequent ventilation of rooms, have been implemented [8,9,10], suggesting that schools do not substantially contribute to the increased circulation of SARS-CoV-2 among local communities.

However, the actual risk and the risk perception of teachers may differ significantly. With the multiwave pandemic dynamics in winter and spring 2020/21 hitting most countries hard, and with health care systems reaching their limits and schools remaining closed, we wanted to better understand teachers’ fear of infection and risk perception. We invited teachers in Germany to participate in the newly developed ImpfREAD survey, in which we asked teachers about their fears of contracting SARS-CoV-2, their intention to be vaccinated against COVID-19, and their opinions on new virus variants and COVID-19-fighting strategies. Demographic and medical parameters were also collected to assess the risk of suffering severe COVID-19. The survey was conducted once and online.

## 2. Materials and Methods

### 2.1. Study Design

We conducted an anonymous, cross-sectional, real-world survey to determine teachers’ fears of contracting SARS-CoV-2, their risk perception, their intention to receive COVID-19 vaccinations, and their opinions on COVID-19 precautions. The survey was conducted nationwide in Germany. Data collection took place between 2 March and 30 April 2021. The online survey was permanently accessible throughout this time period.

### 2.2. Participants and Enrollment

Participants were recruited through the Education and Training Association (VBE, Verband Bildung und Erziehung), German Teachers’ Association (DL, Deutscher Lehrerverband), Union for Education and Training (GEW, Gewerkschaft für Erziehung und Bildung), Association of German Private School Associations (VDP, Verband Deutscher Privatschulverbände e.V., Berlin, Germany), Association of Waldorf Schools (Bund der Freien Waldorfschulen), and the Montessori Association (Montessori Dachverband, Berlin, Germany). The link to the survey was distributed to association members via email distribution lists. The educational network News4teachers website featured the survey. Directly contacted school boards distributed the link via distribution lists. The Institute for Social Medicine, Epidemiology and Health Economics’ Twitter account (@CSozialmedizin) posted the survey in a newsfeed. A snowballing system was encouraged, asking participants to share the survey with other teachers. A total of 6995 individuals attempted to complete all or part of the survey.

The study was conducted in accordance with the ethical standards of the 1964 Declaration of Helsinki. The survey was reviewed by the Berlin Ethics Committee. Since the data in our study were collected anonymously (no code lists are available, and no connections can be made between the collected data and personal data), ethical approval was waived [11]. The data collection took place in accordance with the European Data Protection Regulation. Prior to participating in the survey, participants were given a description and objectives of the study. Written informed consent was obtained from all the participants prior to study enrollment. The study was approved by the institutional data protection board and quality management (Clinical Study Center (CSC) and Clinical Trial Office (CTO) of Charité—Universitätsmedizin Berlin) based on the given data protection concept.

### 2.3. Survey and Outcomes

The survey was newly developed in a multi-disciplinary research collaboration between life scientists, psychologists, and physicians (Appendix A). The survey was pre-tested by 15 independent volunteers, including teachers. The primary outcome was the impact of the COVID-19 outbreak on teachers’ individual health-related worries and their intentions to receive COVID-19 vaccinations. The survey consisted of 92 items capturing 9 themes: (a) socio-demographic characteristics, such as age, gender, and school type, as well as teachers’ fear of contracting SARS-CoV-2, (b) teachers’ intentions to get vaccinated, (c) prioritization of vaccine distribution and school re-openings, (d) teachers’ fear of suffering severe COVID-19, (e) teachers’ opinion on compulsory and (f) influenza vaccinations, (g) teachers’ concerns about new variants and the third COVID-19 wave, (h) teachers’ conspiracy mentality, and (i) socio-demographic characteristics, such as numbers of children and underlying diseases. Depending on the item, participants could either indicate how likely they agree to each item on a five-point Likert scale, ranging from “I agree absolutely” to “I don’t agree at all”, answer “yes” or “no”, or choose from given answers.

### 2.4. Data Collection

Data collection was performed by means of a structured, anonymous, self-administered questionnaire using the Research Electronic Data Capture (REDCap) data management platform version 10.6.14 hosted at Charité—Universitätsmedizin Berlin. REDCap was created in 2004 at the Vanderbilt University, US and is a secure web application for building and managing online surveys and databases [12].

### 2.5. Statistical Analyses

Continuous variables were described as the median with interquartile range (IQR); categorical variables were summarized as percentages. Univariate data were analyzed using Microsoft Excel 2000 and GraphPad Prism version 9.0.0 (San Diego, CA, USA). To address potential sources of bias, a multivariable regression analysis was performed using R software (Version 3.6.1, R Development Core Team, Vienna, Austria) with a dichotomic outcome for anxiousness of getting COVID-19 (yes/no), adjusting for age (in years), gender (male, female), re-opening of schools is the highest priority (I absolutely agree, I agree, neither nor, I don’t agree, I don’t agree at all), risk perception of getting COVID-10 (high, moderate, little, no), teaching a foreign language (yes, no), intention to get vaccinated (yes, no, not sure), schools for children with special needs (yes, no), and Waldorf schools (yes, no). All *p*-values  <  0.05 were considered to be significant.

## 3. Results

### 3.1. Participants

The online survey was accessed by 6995 individuals. A total of 242 individuals either did not give informed consent or did not provide any information. A remaining 6753 teachers who provided their informed consent were included in the study and their responses were analyzed. Most of the participants (42%) were recruited through the educational magazine News4teachers; others were recruited via VBE and GEW (33%), school boards (17%), and other sources (8%) (Appendix A).

### 3.2. Demographic Data

The study was conducted with 6753 teachers in Germany. The geographical distribution of the participating teachers was rather homogeneous, with most participants coming from northern Germany (Appendix A). Table 1 lists the socio-demographic characteristics of the participants. The median age was 43 years (IQR: 36–51 years) and 77% were female. Most participants worked in high schools (29%) and elementary schools (26%), and 7% taught in private school settings (Waldorf schools, Montessori schools, and other private schools). Most of the participating teachers (40%) taught students in grades 7–13, and German language (49%) and mathematics (42%) were the main subjects. Many of the teachers (64%) had children; 48% had two children, 22% had three or more children, and 13% were single parents. Almost 5% of the teachers had had COVID-19 and 16% had been vaccinated against COVID-19. A significant portion (29%) had one or more pre-existing conditions, for example, 18% had asthma, another respiratory disease, or hypertension, and 2% were obese.

### 3.3. Teachers’ Fears of Contracting SARS-CoV-2 at School

A striking majority of the participating teachers (73%) feared contracting SARS-CoV-2 at work (Figure 1, Appendix A). Only 11% of participants were not at all afraid of getting infected. Students (98%), younger (44%), and older colleagues (30%) were named as major threats of infection (Figure 1, Appendix A). Interestingly, 77% reported that teachers also posed a risk to students (Appendix A). A majority (71%) feared that students might carry and transmit SARS-CoV-2 even if asymptomatic (Figure 1, Appendix A). Additionally, many teachers agreed that schools and students substantially contribute to spreading SARS-CoV-2 in communities and schools (66% and 65%, respectively, Figure 1, Appendix A).

Multivariable regression analysis adjusting for age, gender, intention to get vaccinated, school types, the opinion on whether opening schools is the highest priority, the risk perception of getting COVID-19, and the class subjects taught, revealed that female teachers (OR 1.92, *p* < 0.001) had significantly higher odds of being more anxious about getting COVID-19 when compared to their male colleagues (Figure 2, Appendix A). In contrast, teachers who do not want to or are unsure about getting vaccinated had significantly lower odds of being anxious about getting infected (OR 0.02, *p* < 0.001, OR 0.17, *p* < 0.001, respectively) compared to teachers intending to get vaccinated. In addition, teachers from Waldorf schools and schools for children with special needs had significantly lower odds of being anxious about getting COVID-10 (OR 0.38, *p* < 0.001; OR 0.63, *p* = 0.0264, respectively). Teachers who perceived their risk of getting COVID-19 as moderate, low, or not existing had significantly lower odds of being more anxious (OR 0.12, *p* <0.001; OR 0.03, *p* < 0.001; OR 0.01, *p* < 0.001, respectively) compared to teachers who perceived their risk as high (Figure 2, Appendix A). Strikingly, teachers who opposed the re-opening of schools had significantly higher odds of being more anxious compared to teachers who considered school re-openings very important (OR 8.94, *p* < 0.001); specifically, the probability of being more anxious was 9 times higher. At last, teachers of foreign languages had significantly lower odds of being more anxious compared to teachers not teaching foreign languages. With a tendency towards significance, music teachers showed higher odds of being anxious compared to non-music teachers (OR 1.34, *p* = 0.050). Overall, the probability of being anxious about getting COVID-19 significantly decreased with age, with the odds decreasing by 0.98 times per year of year, that is, younger teachers had significantly higher odds of being more anxious compared to older teachers (*p* < 0.001).

Furthermore, we asked about possible conflicts with parents’ and teachers’ opinions on COVID-19 precautions. A majority (47%) felt that parents were too careless about SARS-CoV-2 and thought primarily of themselves and childcare (45%, Appendix A). Most teachers (58%) reported feeling under pressure to implement COVID-19 precaution state guidelines. Similarly, 49% felt they did not meet the expectations of parents and the states regarding the implementation of COVID-19 precautions while educating in a child-friendly manner and to a high standard (Appendix A). Many teachers (45%) had given in at some point to parents’ insistence on additional COVID-19 precautions (Appendix A).

### 3.4. Teachers’ Fear Suffering Severe COVID-19 and New SARS-CoV-2 Variants

A total of 35% of the participating teachers considered their risk of suffering severe COVID-19 as being increased, high, or very high (Table 2). Many teachers volunteered to be tested for SARS-CoV-2 infections depending on the risk situation or at least once a week (42% and 33% respectively, Appendix A).

By the end of 2020, new variants of SARS-CoV-2 had spread from abroad to many European countries. These new variants increased concerns about getting COVID-19 in 72% of the participating teachers (Table 2). A majority foresaw the third COVID-19 wave in Germany and had concerns that a strained health care system would limit their ability to access a hospital and, if necessary, an intensive care unit (76% and 65%, respectively). Despite strict lockdown measures in place, 37% of the teachers rated the general non-pharmacological precautions as still too low (Table 2). A striking majority (78%) believed that COVID-19 can also become a seasonal illness due to mutations in SARS-CoV-2 (Appendix A).

### 3.5. Teachers’ Opinions on Vaccinations and Priority Settings

At the time of the survey, only a minor fraction of the participating teachers (16%) had received a COVID-19 vaccination (Table 1). However, most teachers (77%) intended to get vaccinated (Figure 1). Only 6% were unsure about the COVID-19 vaccination. Many teachers (72%) showed a strong preference for the BioNTech vaccine (Appendix A). A total of 43% and 22% of the participating teachers would also choose the Moderna and AstraZeneca vaccines, respectively. Surprisingly, 23% of the teachers did not have any preference towards a specific vaccine (Appendix A). The main reasons given by most of the participants to get vaccinated were self-protection against getting COVID-19 (97%) and protection for family, friends, children, and others (92%, 75%, and 78%, respectively) (Appendix A). However, only 26% of the teachers chose returning to school quickly as the reason to get vaccinated. On the other hand, some teachers refrained from COVID-19 vaccinations for various reasons, such as unknown long-term effects (87%), vaccines not being adequately tested (78%), and concerns about adverse events (76%). Many participants felt a distrust of the manufacturers and vaccines (49% and 52%, respectively) (Appendix A). Surprisingly, many teachers (67%) supported the notion that all colleagues should be vaccinated against COVID-19 before returning to work (Appendix A).

In Germany, there are no compulsory vaccinations. However, in our study, teachers voted rather equally for and against a compulsory vaccination against COVID-19 discussed by policymakers (43% and 40%, respectively) (Appendix A). Surprisingly, a majority (49%) supported the idea of compulsory vaccinations for other infectious diseases (Appendix A).

A large majority (70%) reported that they regularly checked that they have received all recommended vaccinations. In addition, almost 62% considered the recommendations of the Standing Committee on Vaccination (STIKO) to be appropriate (Appendix A).

Until June 2021, the COVID-19 vaccine allocation was subject to being subdivided into priority groups by law in Germany. Of the participating teachers, 71% agreed with this division (Appendix A). Similarly, 73% agreed that teachers should receive the vaccination earlier, when priority groups were changed in late February 2021. Furthermore, a majority (53%) rated the re-opening of schools as being a top priority (Appendix A). However, regarding new virus variants and a third COVID-19 wave, more teachers (76%) wanted to proceed cautiously in re-opening schools (Table 2).

## 4. Discussion

With the second COVID-19 wave hitting Germany at the end of 2020, schools became the focus of attention. Vaccinations had just begun, but although teachers’ vaccination intention was above average and their role in society is highly acknowledged, they were not prioritized for vaccinations. A public debate arose around teachers’ vaccinations and the role of schools and children in spreading SARS-CoV-2. To investigate factors associated with teachers’ intentions to be vaccinated against COVID-19, we conducted a cross-sectional survey of 6753 teachers in Germany. Using the newly developed ImpfREAD survey, we aimed to investigate both fears and risk perceptions regarding a SARS-CoV-2 infection at school, as well as teachers’ attitudes towards vaccinations.

So far, only a few studies have investigated teachers’ fears of contracting SARS-CoV-2. A cross-sectional study in summer 2020 in Berlin, Germany revealed that about half of the school staff showed a medium to a very strong fear of infection, and 59% reported a moderate to a very high perceived risk of infection [13]. However, this study comprised only 112 teachers at 24 schools in Berlin, mirroring only a limited number of teachers’ opinions, which can largely depend on their respective schools. Another study of almost 1700 teachers was conducted in September 2020 in Spain and reported that nearly half of the teachers surveyed had anxiety [14]. However, both latter studies used anxiety scales, which do not specifically focus on the situation of a pandemic. In our study, we found 73% of teachers were afraid of getting infected with SARS-CoV-2 at school. Female teachers had significantly higher odds of being more anxious compared to their male colleagues, which is consistent with the fact that women are twice as likely to be diagnosed with an anxiety disorder than men [15]. In addition, women have unique healthcare needs, which are reflected by their higher medical care service utilization and higher rates of anxiety [16,17]. Further studies are needed to investigate whether women in the overall population are indeed more anxious about getting infected with SARS-CoV-2.

It is striking that teachers’ anxiety correlates with their attitudes towards school re-openings, that is, the lower teachers rated school re-openings as a priority, the greater the odds were of them being anxious. Interestingly, teachers at Waldorf schools had significantly lower odds of being anxious compared to teachers at primary schools. Rudolf Steiner’s concept of health strongly incorporates a salutogenetic regulatory principle of the human organism, which could be the cause of the Waldorf teachers’ reduced anxiety.

In our study, 4.6% of the teachers had contracted COVID-19, which is similar to the rate of the overall population [1]. A recent study showed that teachers do not bear an increased risk of contracting SARS-CoV-2 [18], however, the perceived risk may differ significantly. Our survey showed that nearly 35% of the teachers assessed their risk of suffering severe COVID-19 as at least increased, and that a large majority fear not receiving adequate medical care due to the overloaded health system. Many patients who were hospitalized with severe COVID-19 have an underlying disease, are older, or are very obese [19,20].

In our study, the probability of fear was highest among the youngest and decreased with age, which strikingly contradicts the actual risk for severe COVID-19 disease progression. The median age of the teachers was 43 years, with under 9% reporting having asthma or another respiratory disease, suggesting that the teachers in this study do not have a higher risk of severe SARS-CoV-2 infection progression than the overall population. The fear of the youngest may, besides other factors, be anxiety triggered by a more frequent social media use [21], which most likely increased during the pandemic.

It can also be speculated that teachers only perceive their risk as being high. In fact, our data show that teachers who assessed their risk as high or moderate had significantly higher odds of being more anxious compared to their colleagues who perceived their risk as low, respectively. Further studies are needed to evaluate the reasons why anxiety promotes teachers’ risk perception.

Although it has been discussed that schools may be COVID-19 infection hotspots, the current evidence suggests against schools having a major role in driving the pandemic. Children and adolescents are typically less susceptible to infection [22,23,24,25] than older individuals. Additionally, previous data showed that children are rarely the index cases of clusters [8,9,22]. In Germany, only 48 COVID-19 outbreaks were registered at schools with ≥2 cases between January and August 2020 [1], accounting for 0.5% of all reported outbreaks. Further studies have concluded that the re-opening of schools in Germany under strict hygiene measures has not increased the number of newly confirmed SARS-CoV-2 infections [26,27,28]. After SARS-CoV-2 infections, children usually show milder symptoms [29,30] and up to 50% of them may remain asymptomatic [31]. The actual number of infected children attending school may, therefore, be higher than anticipated. Therefore, the concern of a large majority of our interviewed teachers that children may attend school asymptomatically despite being infected is justified. On the other hand, child-to-child transmission seems to be lower than intrafamily transmission and transmission from and between adults [32,33]. In addition, larger school outbreaks are rather associated with higher SARS-CoV-2 positivity rates in school staff and secondary transmissions from teachers [34,35]. This is probably why many participating teachers in our study supported the notion that all colleagues should be vaccinated before returning to the classroom.

During the 2020/21 SARS-CoV-2 pandemic, many countries closed schools. Restrictions on large and prolonged social gatherings, such as educational closures, have been identified as one of the most effective non-pharmaceutical interventions [36]. However, school closures may substantially disrupt the lives of children and their families and may have consequences for children’s mental health [37]. In our study, most teachers reported believing that schools substantially contribute to spreading SARS-CoV-2 and are advising to proceed with caution when re-opening schools, despite previous studies convincingly showing that schools play a subordinate role in spreading SARS-CoV-2 [27,28]. We can speculate that teachers at German schools are performing a continuous balancing act between caring about their health and well-being and their pedagogical mission. Indeed, according to the latest findings, teachers emphasize prioritizing and promoting the health and well-being of pupils and teachers in the school learning environment [38].

The mutant line B.1.617, also called the Delta variant, was first detected in October 2020 in India. In Germany, it was identified for the first time in March 2021 [39]. Since then, the Delta variant has been spreading, also among the vaccinated, and has become the predominant variant in Germany since the end of June. The Delta variant is more contagious and transmits easier [40]. Since our study was conducted in March and April 2021, a time when the Delta variant had only recently been identified in Germany, it can be assumed that teachers’ fears and risk perceptions were not influenced by the Delta variant.

So far, there is little scientific data on whether the Delta variant is more likely to cause severe disease than the original strain, and on how adept it is at evading the immune system. Nevertheless, it would be interesting to investigate whether the occurrence of the Delta variant has an influence on the population’s vaccination intentions and to what extent hygiene and protective measures continue to be perceived despite decreasing restrictions.

## 5. Limitations

We wish to highlight that most of the participating teachers were recruited via a news magazine and teachers’ associations. Teachers who were neither members of the associations nor visited the webpage of News4teachers could, therefore, not be included in the study. Furthermore, teachers from vocational schools were not included. Their fears and risk perceptions may differ, as they teach adolescents and young adults. At last, our survey was conducted from March–April 2021. During this dynamic time, Germany faced the second and third COVID-19 waves, vaccination plans changed, schools re-opened and closed depending on the state, and the government relaxed and tightened lockdown measures depending on the numbers of new cases, all of which possibly have an impact on teachers’ fears and risk perception. The proportion of female teachers in our study reflects the national gender proportion of teachers. External validity is, therefore, provided.

## 6. Conclusions

To the authors’ knowledge, this is the first study to describe teachers’ fears and risk perception of COVID-19 as well as their attitudes towards COVID-19 vaccinations in a nationwide survey. In summary, the results of this study show strong fears and risk perceptions among teachers in Germany, which leads to above-average intentions to vaccinate. Fear corresponds with the acceptance of the required COVID-19 protective measures and seems to be a major driver of behavioral attitudes, also dominating rationale. Further studies are needed to evaluate the question of the influence of fear on behavioral modifications towards COVID-19 infection in other occupational groups and the driving gender-dependent factors influencing the development of fear.

## Figures and Tables

**Figure 1 ijerph-18-09218-f001:**
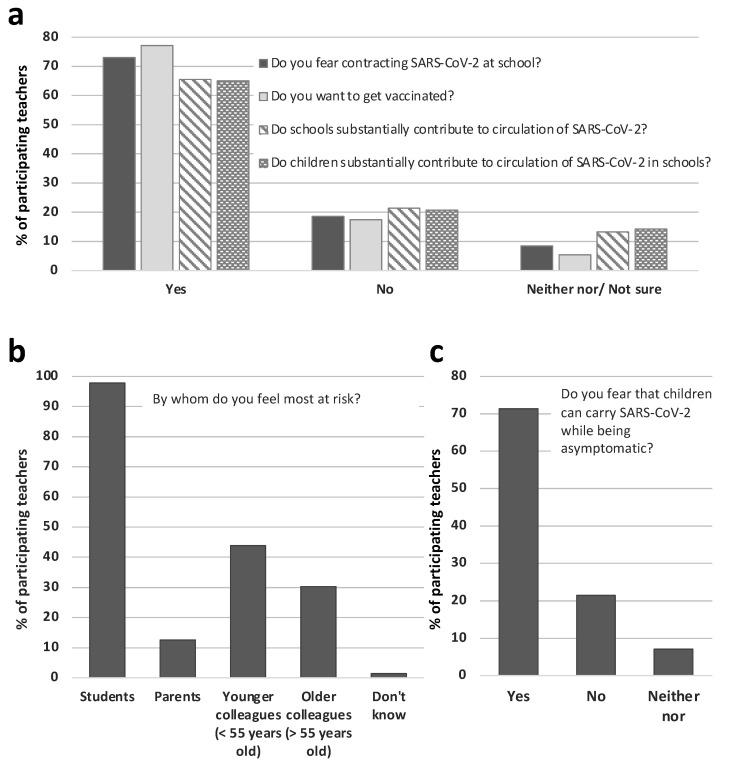
The majority of the participating teachers fear contracting SARS-CoV-2 at school (*n* = 5936), intend to get vaccinated against COVID-19 (*n* = 5850), believe that schools substantially contribute to the circulation of SARS-CoV-2 (*n* = 5936), and believe that children substantially contribute to the circulation of SARS-CoV-2 in schools (*n* = 5936) (**a**). Most teachers fear getting infected from students and younger colleagues (*n* = 4320) (**b**). (**c**) The majority of the participating teachers fear that children can carry SARS-CoV-2 while being asymptomatic (*n* = 5930).

**Figure 2 ijerph-18-09218-f002:**
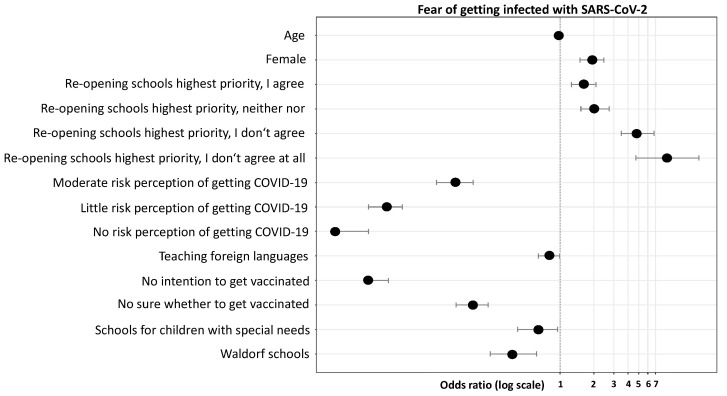
Probability of fear of getting infected with SARS-CoV-2, adjusted multivariable logistic regression analysis. Factors presented are significantly (*p* < 0.05) associated with a reduced probability (left-hand side from the indicated margin) or with an increased probability (right-hand side from the indicated margin) of being anxious about getting infected with SARS-CoV-2.

**Table 1 ijerph-18-09218-t001:** Socio-demographic data of the participating teachers.

Variable	*n* (%)
Gender (*n* = 5894)	
Female	4513 (76.57)
Male	1358 (23.04)
Non-binary/other	23 (0.39)
School types (*n* = 5876)	
Elementary schools	1510 (25.70)
Secondary schools (Hauptschulen) *	212 (3.61)
Secondary schools (Realschulen) *	708 (12.05)
High schools (Gymnasien) *	1729 (29.42)
Comprehensive schools (Gesamtschulen) *	943 (16.05)
Schools for children with special needs	344 (5.85)
Waldorf schools	223 (3.80)
Montessori schools	23 (0.39)
Boarding schools	7 (0.12)
Private schools	151 (2.57)
Language schools	26 (0.44)
Grade levels (*n* = 5525)	
Grades 1–6	1658 (30.01)
Grades 7–13	2191 (39.66)
All grades	1676 (30.33)
School subject (*n* = 5532)	
German language	2735 (49.44)
Mathematics	2316 (41.87)
Science	1490 (26.93)
Science (at elementary schools)	1556 (28.13)
Foreign languages	1708 (30.87)
Sports	1034 (18.69)
Music	874 (15.80)
Others	3235 (58.48)
Age (*n* = 5904)	Median: 43 years
	IQR: 36–51 years
Have children (*n* = 5583)	3596 (64.41)
Number of children (*n* = 2930)	1 child: 1052 (29.37)
	2 children: 1731 (48.32)
	3 children and more: 799 (22.31)
Single parents (*n* = 3497)	462 (13.21)
COVID-19 from recovered (*n* = 5779)	266 (4.60)
COVID-19 vaccine received (*n* = 5878)	949 (16.14)
One or more pre-existing conditions (*n* = 6753)	1929 (29.35)
Asthma and other respiratory diseases	606 (8.97)
Hypertension	584 (8.65)
Obesity	160 (2.37)
Type 1 or 2 diabetes	104 (1.54)
Hashimoto	96 (1.42)
Cancer	71 (1.05)

* In Germany, Hauptschulen, Realschulen, Gesamtschulen, and Gymnasien refer to secondary school forms of intermediate education, that is, level 2 according to UNESCO’s ISCED classification. Students enter these schools after receiving a teacher’s recommendation or, alternatively, after passing an entrance examination.

**Table 2 ijerph-18-09218-t002:** Teacher’s risk perception of suffering severe COVID-19.

Variable	*n* (%)
“How high do you estimate your risk of suffering a severe course of COVID-19?” (*n* = 5784)	
Very little	725 (12.53)
Little	1336 (23.10)
Moderate	1724 (29.81)
Elevated	1230 (21.27)
High	565 (9.77)
Very high	204 (3.53)
**“I am concerned that due to the overload of the health care system, the options to go to a hospital and, if necessary, an ICU, are limited.” (*n* = 5801)**	
I agree completely	1470 (25.34)
I agree	2276 (39.23)
Neither nor	712 (12.27)
I don’t agree	735 (12.67)
I don’t agree at all	608 (10.48)
**“How do you judge the current non-pharmacological precautions (distance + hygiene + mask + warning app + ventilation)?” (*n* = 5829)**	
Exaggerated	1003 (17.21)
Appropriate	2565 (44.00)
Too little	2139 (36.70)
I don’t know	122 (2.09)
**“My concerns about COVID-19 are getting stronger with the new corona mutations.” (*n* = 5726)**	
I agree completely	2111 (36.87)
I agree	2035 (35.54)
Neither nor	381 (6.65)
I don’t agree	531 (9.27)
I don’t agree at all	668 (11.67)
**“One should proceed with caution when re-opening schools, especially in view of the difficult-to-calculate dangers posed by new virus mutations.” (*n* = 5721)**	
I agree completely	2809 (49.10)
I agree	1402 (24.51)
Neither nor	377 (6.59)
I don’t agree	495 (8.65)
I don’t agree at all	638 (11.15)
**“I see a third corona wave approaching unnoticed due to the new variants.” (*n* = 5715)**	
I agree completely	2974 (52.04)
I agree	1386 (24.25)
Neither nor	407 (7.12)
I don’t agree	346 (6.05)
I don’t agree at all	602 (10.53)

## Data Availability

The data presented in this study are openly available in FigShare at https://doi.org/10.6084/m9.figshare.14975286.v1.

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
