# Peer review of "School Teachers’ Self-Reported Fear and Risk Perception during the COVID-19 Pandemic—A Nationwide Survey in Germany"

_ijerph, 2021, doi:10.3390/ijerph18179218_

Round 1

Reviewer 1 Report

Thank you for this interesting piece. The authors focus on a very current and relevant topic and have collected their data timely. The paper is well-written and methodically sound. It is well-documented through the attached supplementary materials.

While the description of the research design is sound, the paper would benefit of more details on when the data was collected and (time frame) and how the questionnaire was pre-tested.

There are some minor spelling issues (e.g. line 69 “theirs”). I would recommend another round of proof-reading and spell-checking.

Author Response

Dear Reviewer 1,

We thank you for your insightful, constructive and positive comments.

We have addressed your points and added more details on the data collection (i.e., from line 81) and on how the questionnaire has been pre-tested (i.e., from line 109).

The paper went for another round of proof-reading and spell-checking.

Kind regards

Reviewer 2 Report

Add a paragraph about the current Delta variant wave and a Paragraph or 2 relating to the Delta variant and how we all are reacting to it.

Author Response

Dear Reviewer 2,

We thank you for your insightful, constructive and positive comments.

We have addressed your point and added 2 paragraphs in the discussion relating to the Delta variant (i.e., from line 351).

Kind regards

Reviewer 3 Report

See the attached for comments and suggestions for the authors. 

Round 2

Reviewer 3 Report

The updates look very good. I enjoyed reading the manuscript.